# Overview of the Circadian Clock in the Hair Follicle Cycle

**DOI:** 10.3390/biom13071068

**Published:** 2023-07-03

**Authors:** Ye Niu, Yujie Wang, Hao Chen, Xiaomei Liu, Jinyu Liu

**Affiliations:** Department of Toxicology, School of Public Health, Jilin University, Changchun 130021, China; niuye22@mails.jlu.edu.cn (Y.N.); haoc20@mails.jlu.edu.cn (H.C.)

**Keywords:** hair follicle cycling, circadian rhythm, metabolism, stem cell, reactive oxygen species

## Abstract

The circadian clock adapts to the light–dark cycle and autonomously generates physiological and metabolic rhythmicity. Its activity depends on the central suprachiasmatic pacemaker. However, it also has an independent function in peripheral tissues such as the liver, adipose tissue, and skin, which integrate environmental signals and energy homeostasis. Hair follicles (HFs) maintain homeostasis through the HF cycle, which depends heavily on HF stem cell self-renewal and the related metabolic reprogramming. Studies have shown that circadian clock dysregulation in HFs perturbs cell cycle progression. Moreover, there is increasing evidence that the circadian clock exerts a significant influence on glucose metabolism, feeding/fasting, stem cell differentiation, and senescence. This suggests that circadian metabolic crosstalk plays an essential role in regulating HF regeneration. An improved understanding of the role of the circadian clock in HFs may facilitate the discovery of new drug targets for hair loss. Therefore, the present review provides a discussion of the relationship between the circadian clock and HF regeneration, mainly from the perspective of HF metabolism, and summarizes the current understanding of the mechanisms by which HFs function.

## 1. Introduction

The circadian clock synchronizes with the day and night cycle around 24 h and is composed of multiple cellular clocks located in organs and tissues. In mammals, the circadian clock affects the expression of multiple genes that control cell metabolism, cell growth and proliferation, immune cell activation, etc. [1,2]. The hair follicle (HF) is a mini-organ that undergoes repeated cycles of regression (catagen), resting (telogen), and growth (anagen), known as the hair cycle [3,4]. The growth cycle is a complex process that requires a variety of HF stem cells and progenitor cells to coordinate the proliferation, differentiation, and migration during HF regeneration [5,6]. Metabolic reprogramming occurs during different HF diet cycles [7]. Peripheral circadian clocks have independent roles that integrate metabolic functions and immune activation [8,9,10,11]. As a peripheral circadian clock, the HF provides a good model for exploring the role of such clocks in the regulation of tissue homeostasis. Plucking HFs is a noninvasive method to measure the circadian gene expression in HF cells [12,13]. It has been reported that basic helix–loop–helix ARNT-like 1 (Bmal1) deficiency can cause a significant delay in anagen progression and increase the accumulation of reactive oxygen species (ROS) in the HF cycle [14]. Furthermore, the circadian clock plays multiple roles in regulating metabolism, aging, and immune responses [15,16,17,18,19]. All of the above are also related to HF regeneration [7,20,21,22,23]. As such, understanding the role of the circadian clock in HFs may facilitate the discovery of new drug targets for preventing hair loss.

## 2. Hair Follicle Cycling

The HF is a tiny but delicate mini-organ derived from ectodermal–mesodermal interactions during the embryonic stage [24]. HFs reside in the dermal layer of the skin [25]. The hair follicle (HF) cycle comprises three phases: anagen, catagen, and telogen [26]. During each growth phase, the HF produces an entire hair shaft from the tip to the root, and during the catagen and telogen phases, the HF stem cells receive a signal from the stem cell niche, begin the next growth phase, and form a new hair shaft. Cyclic HF regeneration is regulated by HF stem cells (HFSCs). Most animals (including humans) exhibit a daily cycle of hair loss and regeneration throughout their life. The HF cycle represents an ideal model for studying tissue regeneration and is related to HFSC quiescence and activation as well as transit-amplifying cell metabolism, differentiation, and apoptosis [3].

## 3. Molecular Circadian Clock in Hair Follicle Cycling

The circadian system comprises a central pacemaker in the suprachiasmatic nucleus (SCN) of the brain, which sets diverse neuronal, endocrine, and behavioral rhythms [27,28]. The circadian clock was long focused on the SCN, which is the central clock. However, extensive evidence indicates that there is a peripheral circadian clock with an independent function [8]. The peripheral circadian clock focuses on organs and tissues with prominent metabolic functions, such as the liver, fat, muscles, and skin. Numerous studies have shown that circadian clock genes play functional roles in human and mouse skin [29,30,31,32]. Lin et al. found that CLOCK-regulated genes are periodically expressed during HF cycling. A genomics approach revealed that CLOCK-regulated genes are significantly upregulated during the telogen phase. Quantitative polymerase chain reaction (q-PCR) experiments also confirmed that the *DBP* gene is upregulated at the transcriptional level, but the protein level of DBP was not evaluated (DBP = D site of albumin promoter (albumin D-box)-binding protein). The researchers also compared the transcription levels of both the telogen and late anagen stages with regard to Dbp, nuclear receptor subfamily 1, group d, member 1 (Nr1d1), period circadian regulator 2 (Per2), Bmal1, and Clock. The q-PCR results showed that the amplitude of the oscillation was significantly enhanced during the telogen stage with regard to Dbp, Per2, and Nr1d1 [14]. Another interesting experiment determined CLOCK, PER1, and BMAL1 protein expression in cultured human HFs. BMAL1 is expressed in the hair matrix, dermal papilla, connective tissue sheath, and outer and inner root sheaths. The researchers did not observe any significant hair-cycle-dependent changes in BMAL1 immunoreactivity. However, PER1 is restricted to the epithelium and exhibits significant changes in expression during the hair cycle. PER1 expression is higher in the catagen phase than in the anagen phase [33]. The authors of another study reported that circadian transcriptomes exhibit distinct gene expression patterns in the telogen and anagen stages of the skin. Specifically, the core clock genes exhibit higher amplitudes of expression in telogen compared to anagen skin [34].

The circadian clock exists in most organisms, and the basic molecular framework of the circadian clock is highly conserved. Circadian clocks are autonomous and impose a period of approximately 24 h, even in the absence of daily environmental signals [2]. The circadian clock controls an organism’s metabolic and physiological functions, which rely on the transcription–translation feedback loop (TTFL) [15]. In mammals, the core molecular oscillator is the TTFL, which involves CLOCK, BMAL1, Periods (PER1, 2, and 3), and cryptochromes (CRY1 and 2). Specifically, the heterodimers of CLOCK and BMAL1 induce the transcription of the *PER1*, *2*, and *3* and *CRY1* and *2* genes that form repressor complexes and bind to CLOCK/BMAL1 to inhibit their transcriptional activity [35,36,37]. Another negative feedback loop reinforces the 24 h of periodicity, the nuclear receptor subfamily 1 group D (REV-ERB), and RAR-related orphan receptor (ROR). Rev-Erb alpha (REV-ERBα), which is also known as nuclear receptor subfamily 1 group D member 1 (NR1D1), negatively regulates the core circadian clock through repression of the transcription factor Bmal1 [38,39,40]. It is also involved in several physiological processes including metabolic and immune functions. TTFL dysfunction of the molecular circadian clock is associated with diseases such as diabetes, obesity, cancer, and immune system dysregulation [41,42].

Akashi et al. utilized human hair follicle cells as an effective model to assess peripheral clocks, providing a noninvasive and straightforward approach. They employed this method to evaluate the rhythmic gene expression in shift workers and found that it could predict the phase of clock gene expression rhythms in a simple and effective manner [12].

## 4. The Circadian Clock Modulates the Hair Follicle Cycle

Recent studies have shown that the circadian clock significantly modulates the HF cycle, both in vitro and in vivo. Knockdown of either BMAL1 or PER1 in cultured human anagen HFs significantly prolongs the anagen phase. Peripheral clocks form an integral component of the human “hair cycle clock” [33]. Lin et al. reported that Clock and Bmal1 play essential roles in regulating the HF cycle. They observed that the anagen progression of HFs was significantly delayed in Clock and Bmal1 mutant mice. Another interesting study has shown that BMAL1 mutant mice exhibited a significant increase in ROS production compared to the control mice [34]. This result also indicated that the circadian clock is involved in modulating cell proliferation and cellular metabolic reprogramming in distinct hair cycle stages [14]. However, further research on the specific mechanisms of the circadian clock in the HF cycle will play a crucial role in the prevention and treatment of tissue regeneration disease.

HFs are constantly exposed to daily fluctuations in temperature, humidity, and light. Blue light may be a direct entrainment signal to skin cells and may disrupt their circadian rhythms at night. When the skin is exposed to blue light, it exhibits increased ROS production, DNA damage, and levels of inflammatory mediators [43]. Clock proteins dominate UVB-induced apoptosis and DNA damage responses. UVB-induced p53 protein accumulation is suppressed by CLOCK depletion, indicating that early steady-state keratinocyte differentiation is triggered by the depletion of either clock gene [44]. Intriguingly, in the absence of light, when Bmal1 is depleted in most SCN neurons, clock rhythmicity is maintained in peripheral tissues. This observation indicates that if the SCN clock is disrupted, non-SCN-derived clocks can synchronize the circadian rhythm of peripheral tissues in constant darkness [45]. ROS levels are elevated in aging *Bmal1*^−/−^ mice, which lack ROS oscillation. However, keratinocyte-specific knockout of Bmal1 does not delay the anagen phase [34]. This suggests that BMAL1 plays a role in regulating this process through the central clock, which is possible through endocrine signaling. This is also possible through non-keratinocyte cell types in the skin, including dermal papillae or preadipocytes [46,47]. In addition, tyrosinase activity and TYRP1/2 expression during the hair anagen phase can be elevated by silencing BMAL1 and/or PER1 in isolated melanocytes [48].

Light can synchronize circadian clocks within the epidermis in the absence of BMAL1-driven clocks in other tissues. A two-branched model for the daily synchronization of tissues can be proposed based on these findings. In the autonomous response branch, circadian clocks are entrained by light without committing to other Bmal1-dependent clocks. In the memory branch, other Bmal1-dependent clocks are used to “remember” the times when external cues are absent [49]. Light stimulation of active HF stem cells through the eyes of animals with prominent hair regeneration occurs via an ipRGC–suprachiasmatic nucleus–sympathetic nervous circuit [50].

The robustness of the circadian clock diminishes with age. Consequently, sleep–wake cycles are disturbed, the capacity to synchronize circadian rhythms in peripheral tissues is reduced, and the circadian clock output is reprogrammed at the molecular function level [51]. Yu et al. identified an miR31–CLOCK–ERK pathway that regulates the aging of HFSCs through trans-epidermal differentiation (miR31 = microRNA 31; ERK = extracellular signal-regulated kinase). They further proved that the premature aging effect induced by miR31 induction or ionizing radiation is ameliorated by the pharmacological inhibition of the MAPK/ERK pathway (MAPK = mitogen-activated protein kinase) [52].

Circadian rhythm sleep disorders are caused by circadian rhythm disruption. However, sleep disorders resulting from physiological and environmental factors can also disrupt normal circadian rhythms [53]. Previous studies have confirmed that sleep disturbances increase the risk of alopecia areata [54]. However, a better understanding of the molecular mechanisms underlying the relationship between the circadian clock and hair loss will contribute to the prevention and treatment of hair loss.

## 5. Metabolic Reprogramming in the Hair Follicle Cycle

Hair follicle regeneration is powered by HF stem cells, which requires the suppression of the metabolic switch from glycolysis to oxidative phosphorylation and glutamine metabolism that occurs during early HFSC lineage progression. Flores et al. demonstrated that HF stem cells utilize glycolytic metabolism and produce significantly more lactate than other epidermal cells. Lactate is essential for HF stem cell activation. Furthermore, the depletion of lactate dehydrogenase (LDHA) results in the failure of HF stem cell activation. Increased lactate production in HF stem cells accelerates their activation and hair cycle progression [55]. Glutamine is an important energy source for HF growth [56]. The authors of one study elucidated the molecular mechanisms underlying HF growth cycle transition, which are regulated by TORC2-Akt signaling-mediated suppression of glutaminase expression and glutamine metabolism (TOR = target of rapamycin; Akt = protein kinase B) [7]. Many studies have confirmed that the mammalian target of rapamycin (mTOR) signaling pathway is essential for regulating the HF cycle [57]. The mTOR pathway is an important regulator of metabolism. HF stem cells and early progenitor cells have distinct metabolic states, and progenitor cells exhibit increased oxidative phosphorylation and tricarboxylic acid cycle activity. Compared to the telogen stage, the early anagen growth phase is associated with increased glutaminolysis and oxidative phosphorylation. HF stem cell differentiation requires mitochondrial metabolism and glutaminolysis. Hughes et al. reported that blocking AKT/mTOR dramatically reversed the age-dependent MSC senescence phenotype, including self-renewal and potential differentiation functions. ROS are required for the regeneration of the HF cycle and skin homeostasis because they promote the proliferation of HF cells [58]. Intrinsic ROS drive HF cycle progression through DNA damage and repair, followed by apoptosis [59]. However, an overdose of ROS may have adverse effects on HFs, leading to hair loss. Consequently, the reduction in ROS accumulation in HFs may be an important strategy for HF repair and regeneration. The inhibition of the AKT/mTOR pathway has been reported to reduce ROS production and subsequent oxidative DNA damage [60]. Furthermore, mTOR complex 1 (mTORC1) signaling coincides with HFSC activation during the telogen-to-anagen transition. mTORC1 signaling negatively affects the bone morphogenetic protein (BMP) signaling pathway, suggesting that mTOR signaling regulates stem cell activation by counterbalancing BMP-mediated repression during hair regeneration [20]. The dermal papilla Wnt/β-actin signal is essential for regulating the activity of keratinocytes and the hair growth phase of the hair cycle (Wnt is a portmanteau of the names Wingless and Int-1) [47]. Fibroblast growth factors and Wnt signaling pathways in the dermal papilla regulate the duration of the hair growth phase. Ablation of fibroblast growth factor signaling in the dermal papilla results in an extended anagen phase via interaction with the Wnt pathway [61]. The Hoxc gene is sufficient to activate dormant dermal papilla niches and promote regional HF regeneration through canonical Wnt signaling [62], and the small GTPase Cdc42 plays a crucial role in skin development and maintenance. Cdc42 controls the differentiation of skin progenitor cells into the HF lineage and regulates catenin turnover [63]. These results suggest that catenin signaling is crucial for promoting HF growth. The hypoxia-inducible factor 1-alpha (HIF-1α) signal plays an important role in the regulation of energy metabolism. Recently, Wang et al. reported that hypoxia microenvironment-active HIF-1α signaling increases glutamine metabolism in keratinocytes and promotes HF regeneration [64].

## 6. The Circadian Clock Controls Metabolism

The circadian clock is tightly coupled with the expression of a large array of metabolic and physiological genes, as demonstrated in the diurnal rhythm [65,66]. Glucose metabolism is an example of the circadian regulation of peripheral metabolism. Dysregulation of the molecular circadian clock is associated with the extended dawn phenomenon in type 2 diabetes, suggesting that the physiological diurnal rhythms of glucose metabolism rely on circadian clock function [67].

### 6.1. Core Circadian Clock Genes and Metabolism

Several studies have demonstrated that HF regeneration is associated with metabolic reprogramming [7,22]. Understanding the role of the circadian clock in metabolic regulation will provide new insights into HF regeneration. BMAL1 knockdown diminishes metabolic rhythms, whereas CRY1 and CRY2 perturbations generally shorten and lengthen the rhythms, respectively [68]. The expression levels of the hepatic glucose transporter GLUT2 and glucokinase (GCK) follow daily rhythms, with peaks coinciding with feeding periods [15]. Inactivation of the known clock components Bmal1 (Mop3) and Clock suppresses diurnal variations in glucose and triglyceride levels. Bmal1-knockout mice exhibit impaired gluconeogenesis, which is reduced in clock mutant mice [69]. *Bmal1*^−/−^ mice exhibit gluconeogenesis dysfunction, as well as changing activity levels and body weight. To determine whether the function of Bmal1 is tissue-specific, the authors of one study determined whether loss of activity levels and body weight can be rescued after muscle-specific expression of Bmal1 in *Bmal1*^−/−^ mice [70]. Similarly, in skeletal-muscle-specific *Bmal1*^−/−^ mice, the mRNA and protein expression levels of muscle glucose transporter (Glut4) were significantly reduced. In addition, the transcript levels of two key rate-limiting enzymes of glycolysis, i.e., hexokinase 2 (Hk2) and phosphofructokinase 1 (Pfk1), were significantly reduced [71]. These results imply that Bmal1 modulates glucose metabolism.

Adipocyte-specific knockout of Bmal1 results in obese mice, with a shift in the diurnal rhythm of food intake, indicating that the adipocyte clock plays a role in the temporal organization of energy regulation [72]. The authors of another interesting study reported that the clock point mutation (ClockΔ19) can form heterodimers with BMAL1 but fails to activate E-box gene transcription. Homozygous ClockΔ19 produces metabolic disorders, including hepatic steatosis, obesity, hypertriglyceridemia, and hyperglycemia, and increases the absorption of lipids throughout the day [35,69,73,74,75]. Bmal1 also plays an essential role in regulating the differentiation potential of embryonic stem cells (ESCs). Loss of Bmal1 function results in a shift from glycolytic to oxidative metabolism through the dysregulation of metabolic gene expression [76]. These results indicate that Bmal1 has a non-canonical circadian function, i.e., to regulate metabolic reprogramming in ESCs.

Wu et al. revealed that the circadian clock gates the hypoxic response, and chromatin immunoprecipitation with sequencing (ChIP-seq) analyses of HIF1A and BMAL1 showed that this hypoxia–clock reciprocal regulation occurs at the genomic level [77]. Another study confirmed that the bidirectional circadian HIF pathway plays a critical role in metabolic adaptation. *Bmal1*^−/−^ mouse embryonic fibroblasts (MEFs) exhibit reduced HIF1a accumulation under hypoxic conditions. Moreover, HIF stabilization can alter the transcription of circadian genes; knockout of the clock repressors CRY1/2 stabilizes HIF1a in response to hypoxia [78]. Adamovich et al. discovered that the oxygen consumption rate follows a daily rhythm and synchronizes the circadian clocks through HIF1a. Low oxygen levels significantly improve jet lag adaptation in mice [79]. Rhythmic immune cell recruitment affects infection outcomes, and the circadian clock modulates immunometabolism. Lee et al. observed that the Bmal1-Hif1a pathway regulates metabolic and inflammatory processes. In myeloid-specific Bmal1 knockout (M-BKO) macrophages, aerobic glycolysis and mitochondrial ROS (mROS) levels increased after M1 stimulation. Bmal1 modulates mitochondrial metabolism through finely tuned Hif-1a activity. Furthermore, M-BKO enhances tumor growth [18]. Bmal1-deficient macrophages reduce the activity of NRF2, leading to the elevated production of the proinflammatory cytokine IL-1β. However, this phenotype can be rescued by pharmacological and genetic activation of NRF2. Additionally, in response to lipopolysaccharides, miR-155 suppresses the transcription and translation of Bmal1, which inhibits the activation of NF-κB [11,80,81].

Nicotinamide adenine dinucleotide (NAD) is an essential coenzyme. It is involved in cellular redox reactions and is a substrate for NAD-dependent enzymes. Nicotinamide phosphoribosyltransferase (NAMPT) provides an essential salvage pathway for the sustained release of NAD+ [82]. The circadian clock drives daily fluctuations in NAMPT levels and consequently cellular NAD+ levels. The sirtuin (SIRT) family of enzymes, which includes histone deacetylases, is involved in metabolic regulation, cancer, and senescence. SIRT1 utilizes NAD+ as a substrate to remove acetyl groups from its target proteins. The circadian clock controls NAMPT-mediated NAD+ biosynthesis via the CLOCK-SIRT1 pathway. SIRT1 is recruited to the NAMPT promoter and contributes to circadian synthesis of its coenzyme [83,84]. It binds to CLOCK-BMAL1 in a circadian manner and promotes the deacetylation and degradation of PER2. The SIRT1-dependent deacetylation of the PER2 protein determines its stability [85]. NAD+ is also used by PARP1 to add poly(ADP) ribose moieties to CLOCK, which increases CLOCK/BMAL1 affinity for DNA, delaying repression by the CRY/PER complex [15,86]. The circadian clock is also involved in the mTOR pathway. Liu et al. found that PER2, a core clock protein, functions as a scaffold that tethers tuberous sclerosis complex 1 (TSC1), raptor, and mTOR to specifically suppress mTORC1 complex activity [87]. BMAL1 is a negative regulator of mTORC1 signaling. BMAL1 deficiency increases mTORC1 activity, both in vivo and in vitro. The suppression of mTOR activity increases the longevity of *Bmal1*^−/−^ mice [88].

### 6.2. Environmental Cues That Regulate Circadian Rhythms and Metabolism

The exposure of animals to environmental stimuli, including artificial light, food intake, and exercise, can remodel rhythmic physiological processes. Artificial light at night disrupts the normal light/dark cycle, resulting in metabolic dysfunction. Many studies have shown that long-term nightshift workers display abnormal metabolic functions and an increased risk of obesity and diabetes [15,89,90]. Disruption of the SCN and peripheral clock has also been observed in mice exposed to constant bright light (LL). In mice, a long-term LL cycle completely abolishes the normal circadian variations in insulin sensitivity and reduces the amplitude of rhythms in the SCN [91]. The authors of one study have reported that mice exposed to artificial light at night become obese and develop hepatic steatosis, which is paralleled by the decreased expression of REV-ERBα and upregulation of its lipogenic targets ATP-citrate lyase and fatty acid synthase in the liver [92]. Exposing mice to alternating dim and bright light (dLL) cycles over the course of 24 h provides a temporal cue that mimics day and night. Unlike LL, dLL has fewer adverse effects on circadian rhythms. Mice exposed to dLL exhibit increased body mass and reduced glucose tolerance; however, their caloric intake and total daily activity are not affected [93]. Exposing zebra finches to dim artificial light (1.5 lux) at night increases their nocturnal locomotor activity but does not significantly alter the expression levels of circadian rhythm genes in their brains or livers. This suggests that their behavior and physiology can be altered by dLL, even though no strong shifts occur in the rhythmic expression of molecular circadian pacemakers [94].

## 7. Conclusions and Perspectives

Hair follicle regeneration is controlled by many physiologic and metabolic processes, especially metabolic reprogramming. At the same time, the majority of metabolic genes show a diurnal expression pattern. An illustration of the relationship between the circadian clock and hair follicle cycle is shown in Figure 1. Furthermore, environmental signals such as temperature, light, and food intake can reset the peripheral clock, which is independent of the SCN. The HF provides a good model for examining peripheral clocks that modulate cell and tissue regeneration. Moreover, recent studies have revealed that clock genes are periodically expressed during HF cycling, and that dysregulation of clock genes results in anagen phase delay [14]. PER1 and BMAL1 knockdown in human HFs significantly prolongs the duration of anagen. BMAL1 is not only expressed in the outer root sheath (ORS) of HFs but is also abundantly expressed in the dermal papilla and connective tissue sheath. Additionally, BMAL1 exhibits prominent expression during the anagen and catagen phases of the HF cycle [33]. The temporal and spatial expression patterns of BMAL1 suggest its involvement in regulating the progression of the HF cycle. The cyclical changes in HFs depend on metabolic reprogramming in different growth stages [7] and are accompanied by alterations in ROS levels [59]. Circadian rhythms play a role in controlling metabolism and ROS homeostasis [95,96,97], and local metabolic changes are crucial for hair follicle cycle transition [98]. Peripheral clocks may potentially influence HF regeneration by regulating the local microenvironment of surrounding tissues. However, the epidermal circadian clock does not have an impact on the HF cycle progression [34]. But the dermis, where the mesenchymal dermal papilla resides, directly participates in regulating the hair follicle cycle [99,100,101,102]. There is currently limited research on whether the dermal layer’s circadian clock directly contributes to hair follicle cycle regulation. The levels of ROS in tissues are closely associated with hair follicle regeneration. Loss of the circadian clock in stem cells can also result in a significant increase in ROS production [76]. Furthermore, normal mitochondrial function cannot be maintained in Bmal1-deficient, M1-activated macrophages [11,18,80,103]. Therefore, mROS production and HIF-1a protein stabilization can be potentiated, possibly increasing inflammatory damage. Intriguingly, we have discovered that the expression of HIF-1a and the abundance of M1 macrophages increase during the catagen phase of HFs. ROS levels are also elevated [59]. Studies indicate that circadian–metabolic crosstalk possibly plays a critical role in modulating the HF cycle [7,15,71,78,98]. Therefore, we speculate that the Bmal1–Hif-1a axis serves as a metabolic switch that may be targeted to control macrophage effector functions during HF regeneration. This raises some interesting questions. Does dysregulation of the circadian rhythm and the metabolic state by external environmental factors affect HF regeneration? What are the underlying mechanisms? Despite our growing knowledge of HF biology and the circadian clock, the molecular mechanisms underlying the HF cycle require further investigation.

## Figures and Tables

**Figure 1 biomolecules-13-01068-f001:**
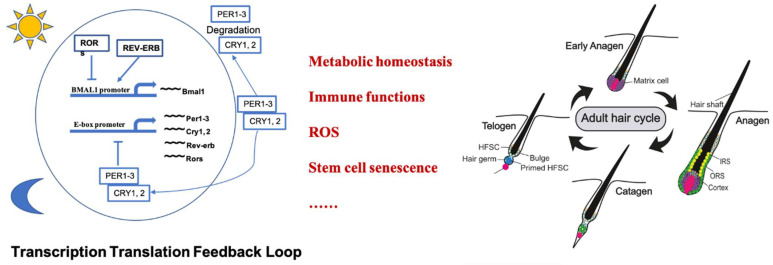
Relationship between the molecular circadian clock and hair cycle. The molecular circadian clock regulates metabolic homeostasis, macrophage activation, ROS homeostasis, stem cell senescence, etc., that significantly affect hair follicle regeneration. Adapted with permission from Ref [5]. published by John Wiley and Sons, 20 December 2020.

## Data Availability

Not applicable.

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
