# Peer review of "Overview of the Circadian Clock in the Hair Follicle Cycle"

_biomolecules, 2023, doi:10.3390/biom13071068_

Round 1

Reviewer 1 Report (Previous Reviewer 1)

The paper is significantly improved and will be relevant and of interest to hair scientists and those working in circadian biology. The attached pdf is annotated with some comments and suggestions. Highlighted words/phrases need minor spelling revisions. 

Please also see a very recent review on the same topic that would be worth referencing.

Liu LP, Li MH, Zheng YW. Hair Follicles as a Critical Model for Monitoring the Circadian Clock. Int J Mol Sci. 2023 Jan 26;24(3):2407. doi: 10.3390/ijms24032407. PMID: 36768730; PMCID: PMC9916850.

Also a study on plucked hair follicles referencing circadian rhythms in shift workers is also relevant

Akashi, Makoto, et al. "Noninvasive method for assessing the human circadian clock using hair follicle cells." Proceedings of the National Academy of Sciences 107.35 (2010): 15643-15648.

The quality of written English is generally very good. Minor changes highlighted in the attached

Author Response

Thank you very much for your very careful review. We have carefully checked the whole manuscript and corrected the typos highlighted in PDF. The changes are highlighted in the revised manuscript. We also properly cited the recommended references.

Reviewer 2 Report (Previous Reviewer 2)

This paper provides a comprehensive review of what is currently known on the role of circadian central clock and its relation with the peripheral hair clock, as well as the regulation of the metabolism.

I do not have special recommendation for changes or edits on the manuscript.

Author Response

Thank you for your positive feedback. We greatly appreciate your careful evaluation of our manuscript.

Reviewer 3 Report (Previous Reviewer 3)

The reviewer considers that the authors now produced a more balanced manuscript and it is acceptable for publication after minor corrections of the reference list.

Citation information is partially missing in references 54 and 55.

There are typos, which may be corrected by the journal.

e.q. page 1, line 39, "As such, Understanding the role of.."

Author Response

We would like to express our gratitude for your time and expertise throughout the review process. I have carefully revised the manuscript accordingly.

This manuscript is a resubmission of an earlier submission. The following is a list of the peer review reports and author responses from that submission.

Round 1

Reviewer 1 Report

A thoughtful and reflective review on this topic is welcome. The authors have captured a wide body of literature and the review covers a very broad range of topics; hair follicle stem cells and regeneration itself is a major undertaking to review.

The organisation of the information could be much improved. The topic is circadian rhythms and I feel this part of the review should come before the information on hair follicles, cycles and regeneration. The biology of the circadian clock should then be related to hair growth and physiology, including stem cells and changes with ageing, which can be mentioned in the text rather than as a whole section. This would then give the review much needed context and relevance to the proposed title. Finally the authors should use the opportunity to identify gaps in our understanding of circadian biology and the hair follicle and to address these in the review rather than at the end of the conclusion. This could be in the form of questions that are subheadings to the review so the reader can appreciate why the information is being included in that section.

The connection between metabolism, circadian clock and hair follicle biology is of interest to the wider readership and is a topic that would seem to draw upon the authors research strengths. This could be the focus of the review.

I would urge the authors to seek out a hair biology specialist to proof read the manuscript before re submission as some statements seem to be rather vague in meaning, clarity or relevance.

Reviewer 2 Report

The review should be more focused on the current degree of knowledge about the hair follicle clock. I think all other portions of the article including the section on structure and growth cycle of the HF is not needed.

In addition the section on the molecular circadian clock and function should be reduced only to give the information that is relevant to understand the principles of the hair clock.

This manuscript needs extensive English improvements.

Reviewer 3 Report

This manuscript summarizes hair follicles and circadian rhythms. The link between the hair cycle and circadian rhythms is an interesting perspective. However, despite the title "Circadian clocks in hair follicle cycles," the majority of the manuscript consists of general information about hair follicles and circadian rhythms, respectively. The connection between the hair cycle and circadian rhythms is limited to a portion of Section 3. Fig. 2, showing the association between the hair cycle and circadian rhythms, is not cited in the text. The title needs to be changed to something appropriate for the content or the content needs to be substantially revised to what the title represents.

Page 4, line 190, the two sentences are exactly the same, only the cited references differ.

Page 4, line 192, there are some grammatical errors, such as capital letters in the middle of sentences. The entire manuscript should be checked.